# Paper Mill Biosolids and Forest-Derived Liming Materials Applied on Cropland: Residual Effects on Soil Properties and Metal Availability

**Bernard Gagnon**  **and Noura Ziadi** *

Agriculture and Agri-Food Canada, Quebec Research and Development Centre, 2560 Hochelaga Boulevard, Québec, QC G1V 2J3, Canada
* Correspondence: noura.ziadi@agr.gc.ca

**Abstract:** Combined paper mill biosolids (PB) and forest-derived liming by-products improve soil properties, but their residual effects following several years of application have hardly been investigated. A 13-year (2009–2021) field study was initiated at Yamachiche, QC, Canada, to assess the residual effects of PB and liming materials on the properties of a loamy soil. The PB was applied during nine consecutive years (2000–2008) at 0, 30, 60, and 90 Mg wet·ha$^{-1}$, whereas the 30 Mg PB·ha$^{-1}$ rate also received one of three liming materials (calcitic lime, lime mud, wood ash) at 3 Mg wet·ha$^{-1}$. No amendment was applied during residual years. Past liming materials continued to increase soil pH but their effect decreased over time; meanwhile, past PB applications caused a low increase in residual soil NO$_3$-N. Soil total C, which represented 40% of added organic C when PB applications ceased, stabilized to 15% after six years. Soil Mehlich-3-extractable contents declined over the thirteen residual years to be not significant for P, K, and Cu, while they reached half the values of the application years for Zn and Cd. Conversely, Mehlich-3 Ca was little affected by time. Therefore, land PB and liming material applications benefited soil properties several years after their cessation.

**Keywords:** liming; organic matter; paper mill biosolids; residual effect; trace metals

## 1. Introduction

Every year in North America, large amounts of paper mill biosolids (PB) from treated effluents (7.1 Mg dry) are generated from the forest industry along with liming materials, such as wood ash (WA) and lime mud (LM) (6.6 Mg dry) [1,2]. Adding these residues to cropland improves soil organic matter content, pH, and major nutrient phyto-availability [3–5]. As long as they respect the local regulatory standards for metal concentrations [4], they can be efficiently and more ecologically used on fields rather than being disposed of in landfill [6].

The application of PB has been largely evaluated for its impact on soil organic matter and N and P contribution to crops in the first year of application and in the subsequent year [3,4,7,8]. However, the residual effects of past PB application on soil properties have been little investigated for a longer period of time, especially after several years of application. Longer-lasting soil organic C improvement has been reported following repeated applications of PB during a few years [9–12], but few studies addressed how long soil C may be improved after PB cessation. Using municipal biosolids, Cogger et al. [13] reported increases in soil total C and Bray-1 extractable P that were maintained for nine years after the end of a 10-year application. They noted, however, a significant decrease in fall NO$_3$-N during the residual period. In contrast, other studies on municipal biosolids reported between 25% and 40% loss in soil organic C within 10 years following the cessation of repeated applications [14–16].

The residual effect of forest-derived liming materials has also been examined, but much more so after a one-time application. Some studies reported no significant change in

soil pH occurring over time with lime mud (LM) and wood ash (WA) three years following a one-time application [17,18]. Moreover, Arshad et al. [17] observed that WA provided 75% more available P than agricultural lime during all the residual years and attributed that to the rise in organic P mineralization, reduced P fixation, and increased P solubility.

The soil bioavailability of metals once material application ceases is a main concern in agriculture and has been part of many studies on municipal biosolids. It was reported that biosolids continue to decompose after cessation and may release the adsorbed metals into the soil solution, particularly in the first five years [16], although the reverse was also observed with the occlusion of metals in Fe-oxides or chemisorption [19]. Biosolid decomposition can also lead to a decrease in soil pH, making metals more available [20]. By contrast, research is yet to be carried out with PB, especially after repeated application and with liming materials. Compared with municipal biosolids, PB mostly represents a low risk for soil metal contamination [21,22] with pollutants rarely exceeding regulatory limits [23,24]. During the years of application, PB adds C and mineral components to soil, thus increasing sorption sites and binding metals in insoluble forms [25]. Despite low inherent metal concentrations, some increases in metal availability in agricultural soils were observed following a fresh addition of PB or deinking paper sludge [26,27], but to our knowledge the residual effects after many consecutive years of application of such products have never been evaluated and would deserve additional study to see any potential impact on soil metal accumulation.

The present study reports the residual effects of the application of PB and forest-derived liming materials on main soil properties and metal availability. We hypothesized that upon material decay and following the cropping season, soil main properties (organic matter, pH, major nutrients) would decrease while metal availability would increase with time. A companion paper that evaluated the effects on plant parameters, including yield and nutrient accumulation, has been previously released [28]. In this study, the liming materials were added to PB with a view to counteract possible soil acidification related to the mineralization of organic N from PB.

## 2. Materials and Methods

### 2.1. Site and Treatment History

The study was carried out at Yamachiche (lat. $46°17'$ N, long. $72°48'$ W, alt. 10 m), Quebec, Canada, on a flat, imperfectly drained Chaloupe loam soil (fine, mixed, frigid Typic Humaquept, according to U.S. soil classification). This study was part of a long-term experiment in which the different materials were applied in early summer at sidedress from 2000 to 2008, and residual effects were followed from 2009 to 2021. The site was under minimum tillage until 2018, including the years of material application, with spring harrowing to prepare the seedbed for annual field crops. In 2019, the site was converted into a perennial hay crop.

The different treatments, plot design, and cultural practices for the years of application have been previously detailed in [29]. Concretely, PB was applied at 0, 30, 60, and 90 wet $Mg·ha^{-1}$, whereas the 30 Mg PB·ha$^{-1}$ rate received also one of the three following liming materials each at 3 wet Mg·ha$^{-1}$: calcitic lime (CL), LM, and WA. Typically, PBs are applied at a rate of 30 Mg wet ha$^{-1}$ year$^{-1}$ in Québec with some additional fertilizer, N, to meet crop requirements [30]. The 60 Mg PB ha$^{-1}$ rate corresponded to rate without N addition, whereas the 90 Mg PB ha$^{-1}$ simulated long-term nutrient accumulation. The experimental layout consisted of plots of $3 \times 10$ m replicated four times in a randomized complete block design.

Material characteristics were reported previously [29] and summarized for the last six years of application (2003–2008) in Table 1. Globally, the PB had a good fertilizing potential with a low metal concentration. For their part, WA and LM had a high total K and Ca content, respectively, but both exceeded the provincial Cd criterion for unrestricted agricultural land use (3.0 mg·kg$^{-1}$). However, these materials were applied at 50% of the mandated provincial limit [31].

**Table 1.** Main chemical characteristics of applied papermill biosolids and liming materials (mean ± standard deviation of 2003–2008 application years).

| Attribute | Units | PB | LM | WA | CL |
|---|---|---|---|---|---|
| pH$_{water}$ | | 5.0 ± 0.9 | 10.6 ± 1.9 | 12.3 ± 0.5 | 9.2 ± 0.5 |
| Moisture | g·kg$^{-1}$ FM | 677 ± 51 | 280 ± 34 | 199 ± 97 | 11 ± 9 |
| Total C | g·kg$^{-1}$ DM | 438 ± 5 | - | - | - |
| Total N | g·kg$^{-1}$ DM | 21.9 ± 5.8 | 0.3 ± 0.3 | 0.3 ± 0.1 | 0.1 ± 0.1 |
| Total P | g·kg$^{-1}$ DM | 3.5 ± 0.8 | 1.6 ± 0.7 | 5.4 ± 1.8 | 0.5 ± 0.2 |
| Total K | g·kg$^{-1}$ DM | 0.6 ± 0.1 | 1.6 ± 2.2 | 20.4 ± 8.0 | 1.9 ± 0.6 |
| Total Ca | g·kg$^{-1}$ DM | 7 ± 2 | 263 ± 41 | 137 ± 26 | 235 ± 24 |
| Total Cu | mg·kg$^{-1}$ DM | 7 ± 3 | 20 ± 10 | 33 ± 8 | 3 ± 0 |
| Total Zn | mg·kg$^{-1}$ DM | 57 ± 20 | 196 ± 42 | 364 ± 41 | 3 ± 1 |
| Total Cd | mg·kg$^{-1}$ DM | 0.7 ± 0.2 | 4.1 ± 0.9 | 5.2 ± 1.2 | 2.0 ± 0.1 |
| Total Mo | mg·kg$^{-1}$ DM | 1.3 ± 0.7 | 0.1 ± 0.1 | 0.0 ± 0.1 | 0.1 ± 0.1 |

PB, paper mill biosolids; LM, lime mud; WA, wood ash; CL, calcitic lime; FM, fresh matter; DM, dry matter.

### 2.2. Field Operations during Residual Years

In residual years (2009–2021), no amendment was applied to the plots except for the mineral N treatment. Annual field crops were grown from 2009 to 2018 and comprised, successively, dry bean (*Phaseolus vulgaris* L.), winter wheat (*Triticum aestivum* L.), grain corn (*Zea mays* L.), malting barley (*Hordeum vulgare* L.), winter wheat, soybean (*Glycine max* (L.) Merr.), winter rye (*Secale cereale* L.), soybean, winter wheat, and spring barley. For 2019 to 2021, a forage mixture of timothy (*Phleum pratense* L.), meadow bromegrass (Bromus biebersteinii Roem. and Schult.), and birdsfoot trefoil (*Lotus corniculatus* L.) was planted as a companion crop with spring barley. For all annual field crops, grains were harvested while straw residues were returned to the field, except for small cereals (wheat, barley, rye) where the straw was sold to other farmers. Forage was harvested three times during each growing season.

### 2.3. Soil Sampling and Analysis

After each crop harvest (the third cut for forage) in residual years (2009–2019), soils were sampled (composite of five cores) to 30 cm depth. Samples were kept moist at 4 °C for pH and NO$_3$-N determination or air-dried and sieved to <2 mm for other soil analysis. The soil pH and NO$_3$-N analysis was performed each year, whereas total C, Mehlich-3-extractable P, K, and Ca, and bio-available metal (Cu, Zn, Cd, Mo) concentrations were measured at a 3 years interval, namely, in 2011, 2014, and 2017. Additional soil sampling was carried out after the first forage cut at the beginning of June 2021 to determine the effect in the course of the 13th year on soil pH and total C, major nutrient (P, K, Ca), and metal (Cu, Zn, Cd) concentrations. For this sampling, Mo availability was not determined since most soil pH ≤ 6.0, which makes Mo barely available to plants [32].

The pH was determined in a 1:2 soil to water ratio. Results from the 10 first residual years (2009–2018) have been reported in [28], and updated here with two more samplings showing a further decrease, where the liming materials reached 0.3–0.6 units over the untreated control for the last sampling (Figure 1).

The NO$_3$-N concentration was extracted with 2.5 g field-moist sample and 20 mL 1 mol·L$^{-1}$ KCl with mechanical agitation for 30 min before filtration [33]. The concentrations were determined through the Cd-Cu reduction method using a continuous-flow injection auto-analyzer (QuickChem 8000 FIA+ analyzer, Lachat Instruments, Loveland, CO, USA). A subsample of dried soil was finely ground to 0.20 mm and then analyzed for total C determination on a LECO TruSpec CN (Leco Inc., St. Joseph, MI, USA).

Soil P, K, Ca, and metal (Cu, Zn, Cd) concentrations were extracted with the Mehlich-3 solution [34]. Metal concentrations (except 2021 samples) were also extracted using DTPA (Cu, Zn, Cd [35]) or water (Mo [36]). Single multielement laboratory Mehlich-3 extraction is relatively simple, cost-effective, and provides generally reproducible results [37].

Concentrations of available P were determined by colorimetry (DU720, Beckman Coulter, Mississauga, ON, Canada) with reaction with the ascorbic acid–molybdate complex [38], whereas concentrations in other elements were determined by the inductively coupled plasma optical emission spectrometer procedure (Optima 4300DV, Perkin Elmer, Shelton, CT, USA).

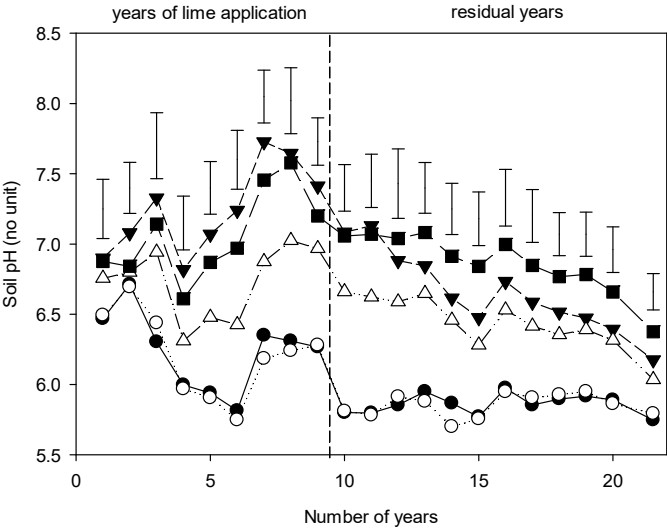

**Figure 1.** Direct and residual effects of liming materials applied during nine consecutive years on the soil pH in the 0 to 30 cm layer. Vertical bars represent the LSD (5%). ● control (0N); ○ combined paper mill biosolids applied alone at 30 Mg wet·ha$^{-1}$ or with 3 Mg wet·ha$^{-1}$ of ▼ lime mud, △ wood ash, or ■ calcitic lime.

### 2.4. Statistical Analysis

All data were first analyzed using the univariate procedure to see their normality and then subjected to a Bartlett's test; Zn (2011, all methods) and Mo (2014) were log-transformed to improve variance homogeneity. Data analysis was performed for separate years to see any effect due to material decomposition using the MIXED procedure of SAS (SAS Studio v.1.21). Main treatment effects were compared using orthogonal linear contrast for PB rate and single degree-of-freedom contrasts for comparison of PB 30 Mg·ha$^{-1}$ alone vs. PB 30 Mg·ha$^{-1}$ + liming, and for CL vs. LM + WA, and LM vs. WA within the PB 30 Mg·ha$^{-1}$ + liming treatments. Statistical significance was set as $p < 0.05$.

## 3. Results and Discussion

### 3.1. Soil Total C

Soil total C continued to benefit from PB after the cessation of land application, but the extent decreased once the C was mineralized (Figure 2). From a respective increase of 3.7, 7.9, and 9.2 g·kg$^{-1}$ compared to the control in fall 2008, it was 2.4, 3.9, and 5.4 g·kg$^{-1}$ in fall 2011, and 1.4, 2.3, and 2.8 g·kg$^{-1}$ at 13 years after the end of application (spring 2021) with the 30, 60, and 90 Mg PB·ha$^{-1}$ rates, respectively. This corresponded roughly to an estimated ~40% of the added organic C being present at the end of continuous application, which fell to 25% after three residual years, and was around 15% thereafter. Based on regression analysis, the model showed a significant decrease in soil C in the first 6 years and then it stabilized at 9.5 years after PB cessation. This indicates that repeated PB application for several years had a long-lasting positive impact on soil total C. It also suggests that a more stable and recalcitrant form could account for the soil total C content some years after PB cessation. Mao et al. [22] estimated that resistant C added with paper waste contributes significantly to the long-term soil-stable C.

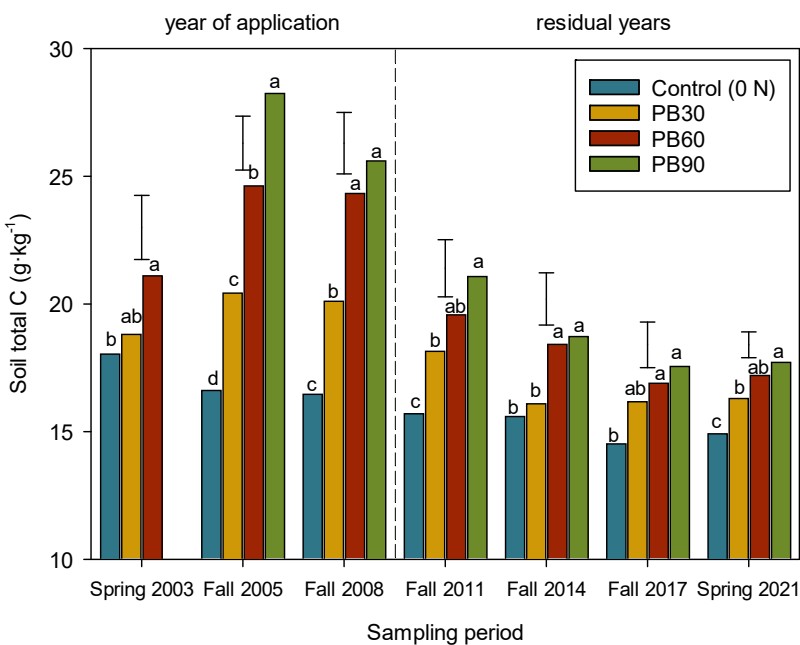

**Figure 2.** Direct and residual effects of paper mill biosolids (PB) application on the soil total C content in the 0 to 30 cm layer. Vertical bars represent the LSD (5%). Different letters within a sampling date represent difference at *p* = 0.05 according to an LSD test. Data for years of application come from [29].

Previous works reported that repeated annual application of PB for three or more years promoted a sustained increase in soil C which was less apparent with a single application [9–12,39]. Cogger et al. [13] applied municipal biosolids continuously for 10 years to a tall fescue (Festuca arundinacea Schreb.), obtaining an increase in soil C equivalent to 28% of material C added, but the increase in soil C was nearly the same (27% of added C) at the end of the nine-year residual period, suggesting soil C stabilization. By contrast, other studies reported between 25% and 40% of soil organic C losses within 10 years after repeated sewage sludge application for 6 to 11 years [14–16]. In this study, by using a linear function of soil total C data versus time after PB cessation at 0, 3, 6, and 9 years for each PB rate treatment (before C stabilization occurring at 9.5 years), we estimated that for the 60 Mg PB·ha$^{-1}$ rate, which summed 70 Mg C·ha$^{-1}$ for the total period of application, net soil C losses reached 0.57 g·kg$^{-1}$ per year (Figure 3).

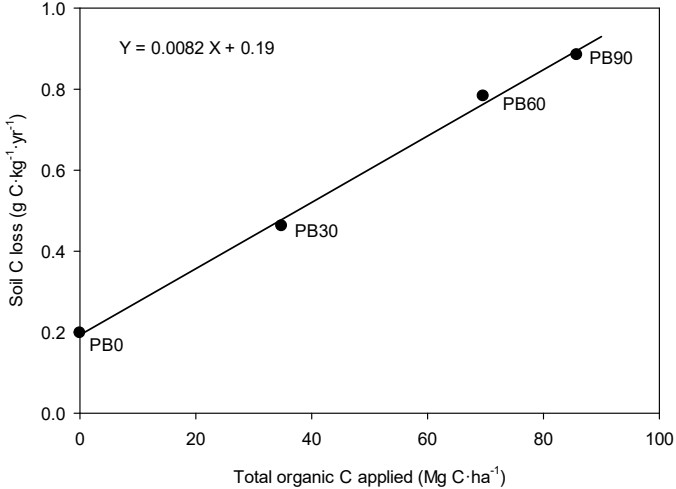

**Figure 3.** Soil total C losses per year from 2008 to 2017 in the paper mill biosolids (PB)-amended plots as a function of the total amount of organic C applied by each treatment in 2000–2008 which totalled, respectively, 0, 35, 70, and 86 Mg·ha$^{-1}$ for each PB rate [29].

### 3.2. Soil NO₃-N

Soil $NO_3$-N concentrations at harvest were increased by the residual PB rate, notably when long-season crops were grown, such as grain corn in 2011 and soybean in 2014, or when grain yields were low, as in 2017 with winter wheat (Figure 4) [28]. However, the increases over the control in those years were much smaller, averaging 2.0 mg·kg⁻¹ for the 90 Mg·ha⁻¹ rate, which were nine times lower that the levels found in the years of PB application [29]. Considering the benefits in plant N accumulation for the first years after cessation [28], it is not expected that PB would cause large environmental $NO_3$-N leaching losses in residual years. Cogger et al. [13] observed a rapid decline in fall $NO_3$-N levels with municipal biosolids, which reached the level of zero-N treatment three years after application cessation.

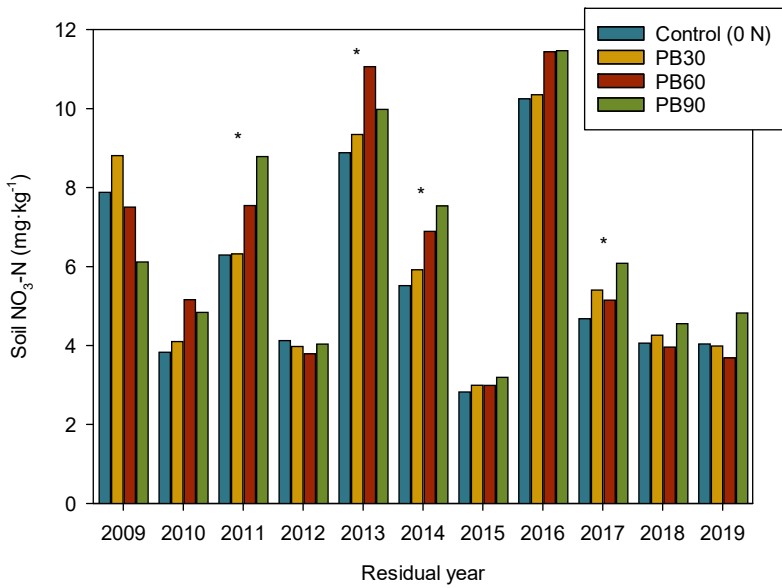

**Figure 4.** Residual effects of paper mill biosolids (PB) application on the soil $NO_3$-N concentrations in the 0 to 30 cm layer at crop harvest. Asterisks indicate significant differences at *p* = 0.05 between the 90 Mg ha⁻¹ rate and the control except in 2013 (60 Mg ha⁻¹ rate vs. control).

### 3.3. Soil Available P, K, and Ca

Soil available P, as determined by the Mehlich-3 extraction, was significantly increased by the PB rate at the third and ninth residual years but not thereafter (Table 2). This increase in soil P translated into higher crop P accumulation up to five years after PB cessation [28]. This meant that PB constituted an efficient source of P for plants and recycled P for a period surpassing the years of application, which has to be considered in fertilizer P recommendations [40]. Cogger et al. [13] found small changes in soil Bray-1 P even nine years after municipal biosolids cessation.

Past liming material application generally increased soil Mehlich-3 P in residual years except 2021 (Table 2). This was related to the increased soil pH, mobilization of P following labile organic carbon added from previous PB, and also a supply of P from WA [3,41,42]. Soil pH exerts an effect on P availability through increasing competition of OH- with $PO_4^{3-}$ ions for binding sites, stimulating microbial activity at neutral pH, and complexing P (precipitation/sorption) with Ca, Fe, and Al ions [43,44].

Soil Mehlich-3 K was not affected by PB in residual years (data not shown) as it was in the years of application [29], indicating that this material was a poor source of K for crops. However, WA increased soil Mehlich-3 K three years after cessation (+43 mg·kg⁻¹ when 30 Mg PB·ha⁻¹ received WA) but not thereafter. This material comprises large concentrations of highly bioavailable K [3,41,45]. Upon crop uptake and likely leaching due to the low binding of K to soil particles, this study indicated that the soil Mehlich-3 K increase was relatively short-lived after cessation of WA [46].

The CL and LM treatments still had significant effects on soil Mehlich-3 Ca in residual years, followed by WA (Table 2). The high contribution of added Ca from CL and LM to soil Mehlich-3 Ca was reported during incubation (82% and 97%, respectively), whereas it was 68% for WA [41]. Despite PB decay and without addition of fresh amendment, our study showed a stability in Ca availability in residual years.

**Table 2.** Effect of residual application of paper mill biosolids and alkaline materials on the soil Mehlich-3-extractable P and Ca in the 0 to 30 cm layer.

| Treatment | Mehlich-3 P (mg·kg$^{-1}$) | | | | Mehlich-3 Ca (mg·kg$^{-1}$) | | | |
|---|---|---|---|---|---|---|---|---|
| | **2011** | **2014** | **2017** | **2021** | **2011** | **2014** | **2017** | **2021** |
| Control (0N) | 69 b | 75 b | 66 c | 65 a | 1095 d | 1256 c | 1097 c | 1123 c |
| PB 30 Mg·ha$^{-1}$ | 71 b | 77 b | 68 c | 62 a | 1224 cd | 1302 c | 1161 c | 1230 bc |
| PB 60 Mg·ha$^{-1}$ | 92 ab | 81 ab | 75 bc | 69 a | 1279 cd | 1430 c | 1285 bc | 1384 bc |
| PB 90 Mg·ha$^{-1}$ | 101 a | 94 ab | 97 ab | 78 a | 1251 cd | 1403 c | 1275 bc | 1283 bc |
| PB 30 Mg + 3 Mg LM ha$^{-1}$ | 102 a | 113 a | 107 a | 89 a | 1738 ab | 1912 ab | 1710 a | 1738 a |
| PB 30 Mg + 3 Mg WA ha$^{-1}$ | 94 ab | 86 ab | 85 abc | 68 a | 1481 bc | 1597 bc | 1422 b | 1393 b |
| PB 30 Mg + 3 Mg CL ha$^{-1}$ | 90 ab | 103 ab | 91 abc | 83 a | 1788 a | 1995 a | 1774 a | 1813 a |
| LSD (5%) | 26 | 32 | 26 | 28 | 273 | 318 | 230 | 242 |
| **Treatment** | Statistical analysis (*F*-value) | | | | | | | |
| | 2.1 | 1.5 | 2.6 * | 0.9 | 6.9 *** | 6.0 *** | 9.2 *** | 8.0 *** |
| **Contrasts** | | | | | | | | |
| PB—linear | 7.7 * | 1.5 | 5.3 * | 0.9 | 1.4 | 1.2 | 3.0 | 2.6 |
| PB vs. PB + liming | 6.0 * | 3.6 | 6.3 * | 2.7 | 17.4 *** | 18.3 *** | 27.8 *** | 19.5 *** |
| CL vs. LM + WA | 0.6 | 0.1 | 0.2 | 0.1 | 2.5 | 3.3 | 4.8 * | 6.1 * |
| LM vs. WA | 0.4 | 3.0 | 3.1 | 2.6 | 3.8 | 4.3 | 6.8 * | 8.9 ** |

PB, paper mill biosolids; LM, lime mud; WA, wood ash; CL, calcitic lime. The years 2011, 2014, 2017, and 2021 correspond to 3, 6, 9, and 13 years after the end of material application. Statistical significance at 5%, 1%, and 0.1% is denoted by *, **, and ***, respectively. Means within a column followed by the same lowercase letter are not statistically significant at *p* = 0.05 according to an LSD test.

### 3.4. Soil Metal Availability

In the years of application, four metals (Cu, Zn, Cd, and Mo) were closely followed, considering their total concentrations in the materials [26]. Copper was the least affected metal. Although no significant effect was detected during the years of application [26], the past PB rate induced a linear effect on Mehlich-3-extractable Cu in 2011 and 2014 and on DTPA Cu in 2014 (Table 3). This meant that Cu was released into the soil in residual years with PB decomposition, which can be attributed to Cu being most preferentially sorbed to organic matter [47].

Soil Zn availability was largely increased by the PB rate in all residual years (Table 4). Soil Zn further increased three years after the last repeated application (fall 2011) and then sharply declined in the sixth year (fall 2014), with fewer variations thereafter (Figure 5A). Soil Cd availability was also increased by the PB rate in residual years (Table 5), but unlike Zn, it decreased soon after application ended and subsequently maintained a fairly constant level throughout those years (Figure 5B). At the time of PB cessation (fall 2008), mean increases in soil available Zn and Cd represented, respectively, 76 and 92% for Mehlich-3 extraction and 55 and 107% for DTPA extraction of the total amounts of each metal added during the years of application. These percentages decreased in residual years to constitute around half the values found in the final application year at the last sampling (36 and 49% for Mehlich-3 and 29 and 67% for DTPA). During the time of application, PB and soil interact to form sorbing sites that retain metals in the soil [25]. Once PB application ceases, PB organic C continues to mineralize at a slower rate, which releases adsorbed metals into the soil solution, but metals may also react with oxides (Fe, Al, Mn) in PB and soil, rendering them less soluble and then less available to plants [14,48]. Trace metals sorbed to oxide surfaces would remain sequestered for a longer period than those complexed by organic C [49]. In this study, metals such as Zn were more

available to plants in the earlier years after PB cessation than during the time of application, likely due to the decomposition of PB, which was still quite significant (Figure 2) [50], and the mobilization of metals by the dissolved organic C produced [51]. Organically bound Zn fraction represented a significant pool in the PB-amended soil, contrarily to Cd [47]. Afterwards, the soil Zn availability was reduced, sorbed to inorganic and very recalcitrant organic components in the soil–residual biosolids mixture [49].

**Table 3.** Effect of residual application of paper mill biosolids and alkaline materials on the soil-available copper in the 0 to 30 cm layer.

| Treatment | Mehlich-3 Cu (mg·kg$^{-1}$) | | | | DTPA Cu (mg·kg$^{-1}$) | | |
|---|---|---|---|---|---|---|---|
| | 2011 | 2014 | 2017 | 2021 | 2011 | 2014 | 2017 |
| Control (0N) | 3.1 c | 4.7 bc | 5.2 a | 4.5 a | 7.1 b | 4.7 bc | 5.1 a |
| PB 30 Mg·ha$^{-1}$ | 4.1 abc | 4.5 c | 4.9 a | 4.6 a | 7.7 ab | 4.4 bc | 4.8 a |
| PB 60 Mg·ha$^{-1}$ | 5.0 a | 5.1 bc | 5.4 a | 5.3 a | 9.0 a | 4.7 b | 5.0 a |
| PB 90 Mg·ha$^{-1}$ | 4.3 ab | 5.5 a | 5.5 a | 4.8 a | 8.1 ab | 5.6 a | 5.1 a |
| PB 30 Mg + 3 Mg LM ha$^{-1}$ | 4.1 abc | 5.4 ab | 5.4 a | 5.3 a | 7.6 ab | 4.4 bc | 4.8 a |
| PB 30 Mg + 3 Mg WA ha$^{-1}$ | 4.4 ab | 5.2 bc | 5.4 a | 4.8 a | 8.1 ab | 4.9 b | 4.9 a |
| PB 30 Mg + 3 Mg CL ha$^{-1}$ | 3.3 bc | 4.6 bc | 4.9 a | 4.2 a | 6.9 b | 4.0 c | 4.3 a |
| LSD (5%) | 1.2 | 0.8 | 0.9 | 1.2 | 1.7 | 0.7 | 0.8 |
| Treatment | Statistical analysis (*F*-value) | | | | | | |
| | 2.3 | 2.0 | 0.7 | 0.8 | 1.1 | 4.2 ** | 1.0 |
| Contrasts | | | | | | | |
| PB—linear | 6.5 * | 6.3 * | 1.3 | 0.7 | 2.5 | 10.1 ** | 0.0 |
| PB vs. PB + liming | 0.2 | 3.5 | 1.1 | 0.2 | 0.1 | 0.0 | 0.2 |
| CL vs. LM + WA | 3.4 | 4.0 | 1.8 | 2.8 | 1.7 | 4.5 * | 2.7 |
| LM vs. WA | 0.4 | 0.4 | 0.0 | 0.7 | 0.3 | 2.2 | 0.1 |

PB, paper mill biosolids; LM, lime mud; WA, wood ash; CL, calcitic lime. The years 2011, 2014, 2017, and 2021 correspond to 3, 6, 9, and 13 years after materials application ending. Statistical significance at 5% and 1% denoted by * and **, respectively. Means within a column followed by the same lowercase letter are not statistically significant at *p* = 0.05 according to an LSD test.

**Table 4.** Effect of residual application of paper mill biosolids and alkaline materials on the soil available zinc in the 0 to 30 cm layer.

| Treatment | Mehlich-3 Zn (mg·kg$^{-1}$) | | | | DTPA Zn (mg·kg$^{-1}$) | | |
|---|---|---|---|---|---|---|---|
| | 2011 | 2014 | 2017 | 2021 | 2011 | 2014 | 2017 |
| Control (0N) | 6.4 bc | 6.3 bc | 5.6 bc | 5.2 bc | 4.0 bc | 3.4 bcd | 2.8 bc |
| PB 30 Mg·ha$^{-1}$ | 5.9 bc | 5.9 bc | 5.5 bc | 4.9 bc | 3.9 bc | 3.5 bc | 2.8 bc |
| PB 60 Mg·ha$^{-1}$ | 8.5 ab | 7.1 ab | 7.0 ab | 5.5 bc | 5.6 ab | 4.1 b | 3.6 ab |
| PB 90 Mg·ha$^{-1}$ | 11.2 a | 8.4 a | 7.8 a | 6.9 a | 7.4 a | 5.2 a | 4.0 a |
| PB 30 Mg + 3 Mg LM ha$^{-1}$ | 6.8 bc | 6.9 abc | 6.5 abc | 6.0 ab | 3.6 bc | 3.4 bcd | 2.9 bc |
| PB 30 Mg + 3 Mg WA ha$^{-1}$ | 6.5 bc | 5.8 bc | 5.9 bc | 4.9 bc | 3.8 bc | 3.0 cd | 2.7 c |
| PB 30 Mg + 3 Mg CL ha$^{-1}$ | 5.7 c | 5.5 c | 4.9 c | 4.7 c | 3.0 c | 2.5 d | 2.0 c |
| LSD (5%) | 3.2 | 1.5 | 1.6 | 1.3 | 2.8 | 0.9 | 0.8 |
| Treatment | Statistical analysis (*F*-value) | | | | | | |
| | 3.1 * | 3.5 * | 3.0 * | 2.9 * | 3.7 ** | 7.1 *** | 4.6 ** |
| Contrasts | | | | | | | |
| PB—linear | 12.6 ** | 10.5 ** | 11.3 ** | 8.4 ** | 11.4 ** | 19.6 *** | 12.7 ** |
| PB vs. PB + liming | 0.2 | 0.1 | 0.3 | 0.3 | 0.5 | 2.0 | 0.8 |
| CL vs. LM + WA | 1.0 | 1.7 | 3.7 | 2.1 | 1.5 | 3.3 | 4.7 * |
| LM vs. WA | 0.1 | 2.4 | 0.6 | 3.4 | 0.1 | 0.6 | 0.1 |

PB, paper mill biosolids; LM, lime mud; WA, wood ash; CL, calcitic lime. The years 2011, 2014, 2017, and 2021 correspond to 3, 6, 9, and 13 years after materials application ending. Statistical significance at 5%, 1%, and 0.1% denoted by *, **, and ***, respectively. Means within a column followed by the same lowercase letter are not statistically significant at *p* = 0.05 according to an LSD test.

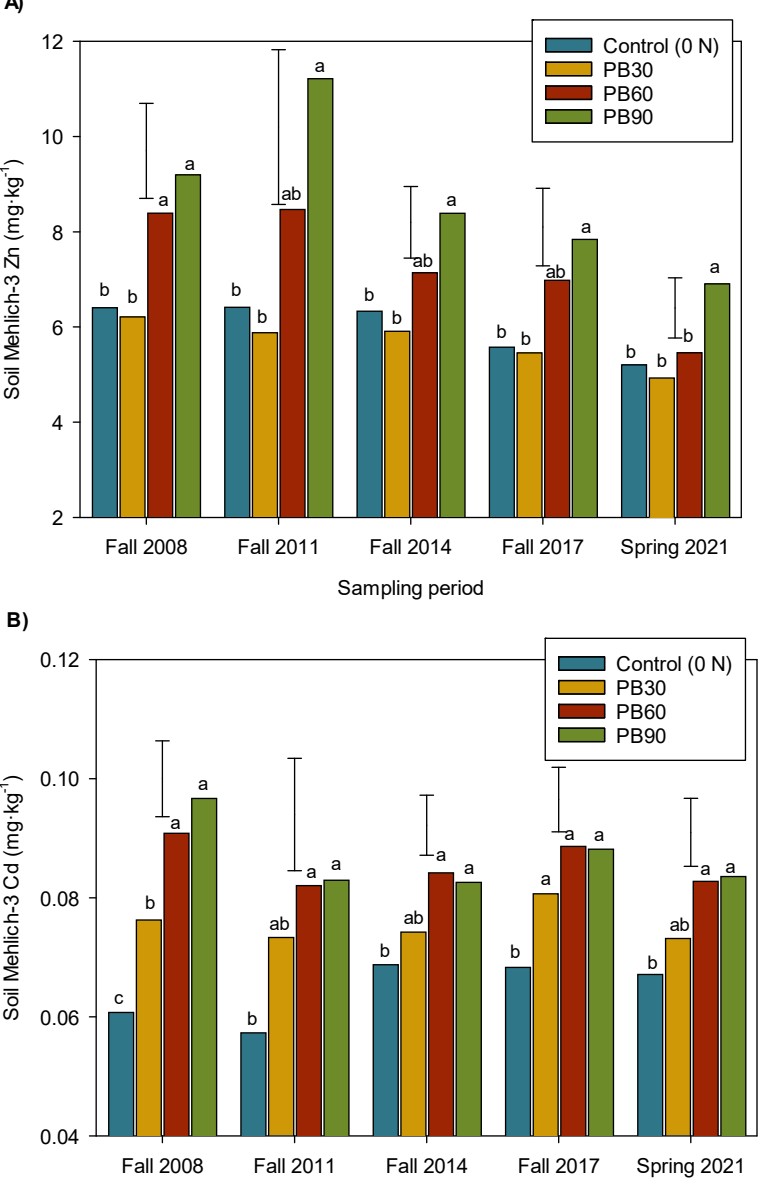

**Figure 5.** Direct (2008) and residual (2011, 2014, 2017, 2021) effects of paper mill biosolids application on the soil Mehlich-3 Zn (**A**) and Cd (**B**) concentrations in the 0 to 30 cm layer. Vertical bars represent the LSD (5%). Different letters within a sampling date represent difference at *p* = 0.05 according to an LSD test. Data for 2008 come from [26].

Past addition of CL decreased the soil-available Cd and, to a lesser extent, the soil-available Zn in residual years (Tables 4 and 5). Their concentrations in the amended soil were similar or slightly lower than those in the unamended control after application cessation. This was related to the long-term positive effect of CL on soil pH (Figure 1) [32], low cumulative metal supply, and lower labile extracted fractions in this treatment [47]. It was reported that the soil pH variations induced by the addition of sewage sludge and liming were a dominant factor in metal solubility [52].

Soil water-soluble Mo responded positively to liming materials in all residual years, but we observed a general decline, from 4 to 6 $\mu g \cdot kg^{-1}$ over the 30 Mg PB $ha^{-1}$ rate applied alone at the last year of application, to 3 to 5 $\mu g \cdot kg^{-1}$ after three residual years, and 1–2 $\mu g \cdot kg^{-1}$ after nine residual years (Figure 6). Even at these low concentrations, increased soil Mo related to past liming addition induced substantial Mo concentrations in soybean grains [28]. For all the duration of the residual years, CL treatment provided

the highest soil Mo concentrations. In this study, a regression analysis indicated that at a pH > 6.2, there is a steady increase in soil water-soluble Mo (Figure 7). O'Connor et al. [53] reported that Mo sorption becomes negligible above pH 6. Moreover, following addition of liming materials, the sorptivity of Mo in acidic soils is partially reversible due to the high solubility of Al-molybdate [54].

Soil water-soluble Mo did not respond to the PB rate in residual years (data not shown) as it did in application years [26]. Previous studies with municipal biosolids, which were richer in Mo than PB, indicated only a small effect on soil Mo sorption due to the presence of Fe and Al oxides but a change in soil pH that can affect Mo retention and release [55].

**Table 5.** Effect of residual application of paper mill biosolids and alkaline materials on the soil available cadmium in the 0 to 30 cm layer.

| Treatment | Mehlich-3 Cd (mg·kg⁻¹) | | | | DTPA Cd (mg· kg⁻¹) | | |
|---|---|---|---|---|---|---|---|
| | **2011** | **2014** | **2017** | **2021** | **2011** | **2014** | **2017** |
| Control (0N) | 0.057 bc | 0.069 c | 0.068 c | 0.067 b | 0.071 bc | 0.061 bc | 0.063 bc |
| PB 30 Mg·ha⁻¹ | 0.073 ab | 0.074 bc | 0.081 ab | 0.073 ab | 0.082 ab | 0.073 ab | 0.075 ab |
| PB 60 Mg·ha⁻¹ | 0.082 a | 0.084 ab | 0.089 a | 0.083 a | 0.090 a | 0.076 a | 0.078 a |
| PB 90 Mg·ha⁻¹ | 0.083 a | 0.083 ab | 0.088 a | 0.084 a | 0.096 a | 0.080 a | 0.083 a |
| PB 30 Mg + 3 Mg LM ha⁻¹ | 0.069 abc | 0.090 a | 0.084 ab | 0.082 a | 0.072 bc | 0.068 ab | 0.071 abc |
| PB 30 Mg + 3 Mg WA ha⁻¹ | 0.070 abc | 0.073 c | 0.082 ab | 0.078 ab | 0.069 bc | 0.062 bc | 0.070 abc |
| PB 30 Mg + 3 Mg CL ha⁻¹ | 0.053 c | 0.065 c | 0.075 bc | 0.076 ab | 0.059 c | 0.054 c | 0.059 c |
| LSD (5%) | 0.019 | 0.010 | 0.011 | 0.011 | 0.017 | 0.013 | 0.013 |
| Treatment | Statistical analysis (*F*-value) | | | | | | |
| | 2.9 * | 6.1 *** | 3.4 * | 2.3 | 4.6 ** | 4.6 ** | 3.1 * |
| Contrasts | | | | | | | |
| PB—linear | 8.9 ** | 11.3 ** | 16.8 *** | 11.6 ** | 10.3 ** | 10.3 ** | 10.2 ** |
| PB vs. PB + liming | 1.6 | 0.3 | 0.0 | 1.5 | 5.0 * | 5.5 * | 2.8 |
| CL vs. LM + WA | 4.5 * | 14.9 *** | 3.1 | 0.6 | 2.5 | 4.8 * | 4.6 * |
| LM vs. WA | 0.0 | 12.6 ** | 0.2 | 0.7 | 0.2 | 1.2 | 0.0 |

PB, paper mill biosolids; LM, lime mud; WA, wood ash; CL, calcitic lime. The years 2011, 2014, 2017, and 2021 correspond to 3, 6, 9, and 13 years after the end of material application. Statistical significance at 5%, 1%, and 0.1% is denoted by *, **, and ***, respectively. Means within a column followed by the same lowercase letter are not statistically significant at *p* = 0.05 according to an LSD test.

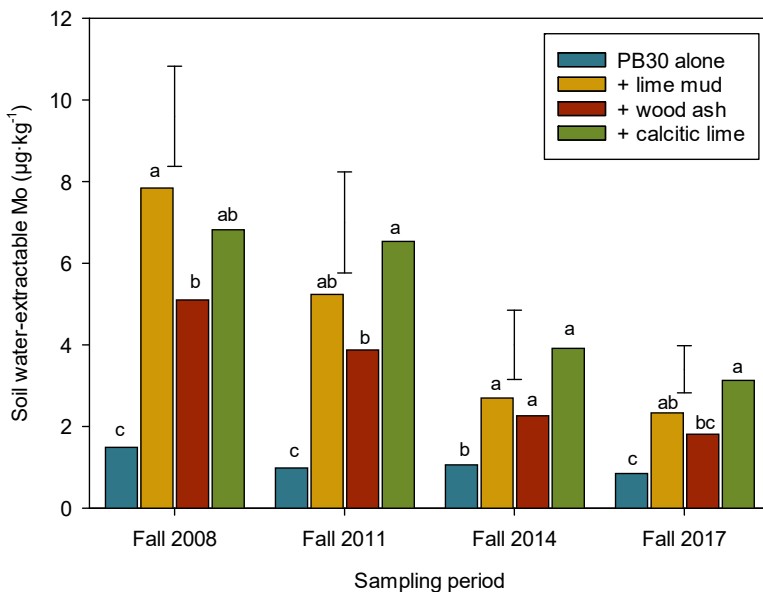

**Figure 6.** Direct (2008) and residual (2011, 2014, 2017) effects of liming materials application on the soil water-extractable Mo concentrations in the 0 to 30 cm layer. Vertical bars represent the LSD (5%). Different letters within a sampling date represent difference at *p* = 0.05 according to an LSD test. Data for 2008 come from [26].

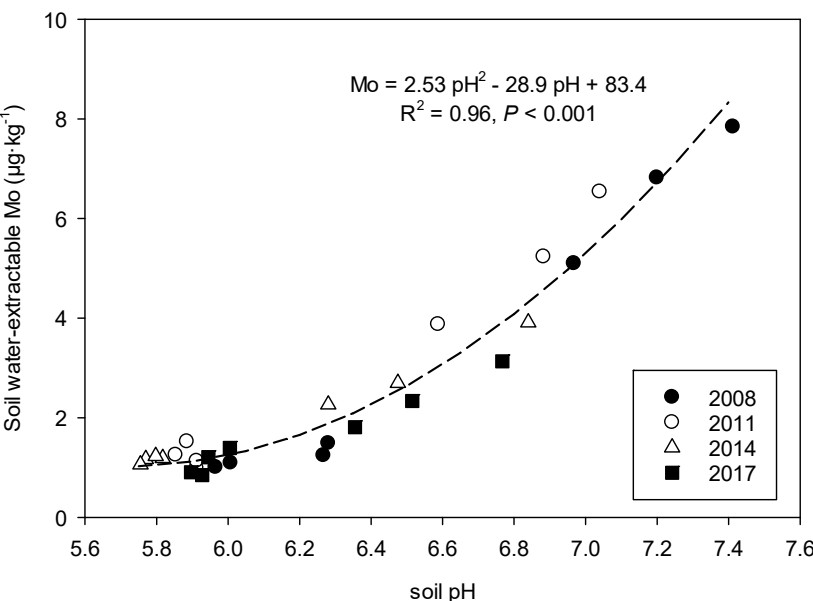

**Figure 7.** Relationship between soil pH and soil water-extractable Mo content for all treatments (PB rates and PB + liming materials) across time samplings.

## 4. Conclusions

This study aimed to determine the residual effects, over a thirteen-year period, of nine consecutive years of application of PB and liming materials on main soil properties and metal availability. During this period, soil pH was higher in past amended CL and LM plots, but gradually declined over time to represent 0.3–0.6 units over the untreated control at the end of the study. The past PB addition continued to improve soil total C, but its effect also declined to stabilize at around 15% of total added organic C after six years. Past PB rate also increased soil $NO_3$-N, but the contribution was negligible, indicating a low risk of leaching. Soil Mehlich-3 P and K decreased in residual years to be not significant at the end of the study, while liming materials continued to positively affect Mehlich-3 Ca. Three years after cessation, the availability of Cu and Zn increased due to PB decomposition and then decreased thereafter. Conversely, the availability of Cd diminished soon after PB cessation. By the end of the residual study, the availability of Zn and Cd reached half the values of those at the end of the material application year, while no more effects were detected for the availability of Cu. For its part, the level of soil water-soluble Mo decreased as the soil pH dropped back towards the control value. Therefore, our study indicated that land application of PB and forest liming by-products promoted soil properties up to 9 years after treatments ceased without increasing the metal availability.

**Author Contributions:** Conceptualization and design of the experiment: N.Z.; methodology, data acquisition, and validation: B.G.; writing—original draft preparation: B.G.; writing—review and editing: N.Z.; funding acquisition: N.Z. All authors have read and agreed to the published version of the manuscript.

**Funding:** This study was funded by Agriculture and Agri-Food Canada's A-base program. AAFC-1555.

**Data Availability Statement:** Data are available upon request.

**Acknowledgments:** We thank Ferme Firmin Fréchette et fils for providing the experimental site for all those years. We are also grateful to Sylvie Côté, Sylvie Michaud, Claude Lévesque, and Gabriel Lévesque for providing technical assistance.

**Conflicts of Interest:** The authors declare no conflict of interest.

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
