# Peer review of "Paper Mill Biosolids and Forest-Derived Liming Materials Applied on Cropland: Residual Effects on Soil Properties and Metal Availability"

_soilsystems, doi:10.3390/soilsystems7020040_

Round 1

Reviewer 1 Report

Moderate English changes required

Reviewer 2 Report

Dear authors,

Please provide a manuscript where all figures can be easily readable. Especially Figures 4, 5 and 6. Also, all parameters in the tables must be explained, especially table 2.

It is not possible to give comments about results that are not shown in the manuscript.

Please correct your manuscript and resend it again.

Best regards 

Reviewer 3 Report

soilsystems-2230633

Paper Mill Biosolids and Forest-derived Liming Materials Applied on Cropland: Residual Effects on Soil Properties and Metal Availability

The manuscript describes residual effects of paper mills bio solids and forest derived liming materials on soil properties and metal availability.

Abstract

The experimental design is not clear or appropriate. How can one check the effect of forest derived liming materials effect if it does not have proper positive control (where only forest derived liming material should be applied)?  How one can see whether the effect was due to liming material or from the PB in a liming + PB 30 MG ha-1 treatment?

What do you mean by little increase, please clarify if it is increase or decrease and whether the observed results were statistically significant or not?

Although authors claim thirteen-year experiment but in the results section of the abstract the did not mentioned what happened after thirteen years on the studied parameters. Please add those in percent increase or decrease term in the abstract.

If availability of Cu and Zn was increased first three years what happened to those parameters in the end of experiment, please add those results. These are important to support your conclusion in the abstract.

I am not sure based on the current experimental setup you can conclude about the effect of liming material on soil properties.  

Introduction

The introduction section is not up to the mark the authors just describes some previous studies on the paper mills bio solids application in the soil and effect on the different soil properties. However, authors fail to describes the research gaps in the manuscript. Why this research is necessary?  Authors themselves describes in the introduction that “PB mostly represent a low risk for soil metal contamination [19-20] with pollutant rarely exceeding regulatory limits” then why there is need to see the residual effects of PB on the heavy metals in the soil. Although decomposition of the biosolids might lead to release those if these are present in the PB. If not as authors claims in the introduction then I could not see any need for measuring residual effect of PB on heavy metals in the soil.

Similarly, in the second paragraph of the introduction section authors describes that “other studies on municipal biosolids reported between 25% and 40% loss in soil organic C within 10 yr following cessation of repeated applications”. So what is novel in this study please clearly elaborate with the addition of some more recent literature and defining the gap in the introduction section.

Materials and Methods

As noted above the objective about the effect of liming material without proper positive control would not be investigated. So the experimental setup has real flaws in it.

Please justify the selection of difference application rates of biosolids in the soil. What was the criteria for this selection?

Statistical analysis seems not clear Bartlett's test used to test homoscedasticity, that is, if multiple samples are from populations with equal variances that is the second assumption of analysis of variance. So if the first criterion to test the normality of the data and second criterion required to test homoscedasticity is fulfilled use ANOVA then which test was applied is not mentioned or discussed in the statistical analysis. So I guess the current statistical analysis is also not correct as well. It seems authors applied analysis of variance but not clearly mentioned it. In this case, parametric ANOVA would not be good example for the low degree of freedom data. It is better to use nonparametric ANOVA to get the fair analysis of the data.

Results and discussion

It is better to separate this into two section 1. Results, 2 Discussion

Then based on your hypotheses you can discuss your results in the discussion section. In the current setting, I could not find the discussion on the hypotheses made in the 9intrdouction section.   

Add statistics analysis in the fig. 2. From the current figure without statistics (please add letter (multiple comparison) to differentiate the differences among the year in C content), I could not see the difference between the years 2003 to 2008 and between 2011 to 2021. Statistics is missing in figs. 4, 5 and 6 as well.

Reviewer 4 Report

Dear authors,

I found it a simple but very interesting work. It is always appreciated to read works carried out in the field and that the monitoring lasts for years.

This type of work demonstrates the effectiveness or ineffectiveness of the amendments applied to the soil, since after so many years the real effect of the amendments on the soil can be observed.

I have added a number of simple comments and suggestions throughout the manuscript.

Best regards

Reviewer 5 Report

1.        Soil organic carbon and nutrient status decreased with the increasing duration of BP and liming material application, but the content of heavy metals increased. Why is this result beneficial to soil parameters (L21)?

2.        No data in Figures 2 and 4-6?

3.        How to calculate the loss of soil organic carbon?
